# Bivalent VSV Vectors Mediate Rapid and Potent Protection from Andes Virus Challenge in Hamsters

**DOI:** 10.3390/v16020279

**Published:** 2024-02-11

**Authors:** Joshua Marceau, David Safronetz, Cynthia Martellaro, Andrea Marzi, Kyle Rosenke, Heinz Feldmann

**Affiliations:** 1Laboratory of Virology, Division of Intramural Research, National Institute for Allergy and Infectious Diseases, National Institutes of Health, Hamilton, MT 59840, USA; josh24344@gmail.com (J.M.); david.safronetz@phac-aspc.gc.ca (D.S.); cmartellaro13@gmail.com (C.M.);; 2Department of Biomedical and Pharmaceutical Sciences, The University of Montana, Missoula, MT 59812, USA

**Keywords:** Orthohantavirus, Andes virus, VSV, vaccine, Syrian hamster, emergency vaccination

## Abstract

Orthohantaviruses may cause hemorrhagic fever with renal syndrome or hantavirus cardiopulmonary syndrome. Andes virus (ANDV) is the only orthohantavirus associated with human–human transmission. Therefore, emergency vaccination would be a valuable public health measure to combat ANDV-derived infection clusters. Here, we utilized a promising vesicular stomatitis virus (VSV)-based vaccine to advance the approach for emergency applications. We compared monovalent and bivalent VSV vectors containing the Ebola virus (EBOV), glycoprotein (GP), and ANDV glycoprotein precursor (GPC) for protective efficacy in pre-, peri- and post-exposure immunization by the intraperitoneal and intranasal routes. Inclusion of the EBOV GP was based on its favorable immune cell targeting and the strong innate responses elicited by the VSV-EBOV vaccine. Our data indicates no difference of ANDV GPC expressing VSV vectors in pre-exposure immunization independent of route, but a potential benefit of the bivalent VSVs following peri- and post-exposure intraperitoneal vaccination.

## 1. Introduction

Hantaviruses are a group of zoonotic pathogens comprising the genus *Orthohantavirus* in the *Hantaviridae* family within the order *Bunyavirales* [1]. They are enveloped particles containing a tri-segmented, single stranded, negative sense RNA genome. The three viral RNA segments are designated small (S), medium (M) and large (L) and encode the nucleocapsid protein (NP), the glycoprotein precursor (GPC), and the viral RNA polymerase (RdRp), respectively [1]. Hantaviruses are rodent or insectivore-borne viruses and primary transmission to humans is thought to largely occur by exposure to excreta and secreta from infected rodents exposure through aerosolization or direct introduction into broken skin or onto mucosal membranes [2]. Hantaviruses are geographically divided into Old World and New World hantaviruses, which are associated with two clinical syndromes in humans—hemorrhagic fever with renal syndrome (HFRS) and hantavirus cardiopulmonary syndrome (HCPS)—respectively [3].

Andes virus (ANDV) is one of the most virulent New World hantaviruses with a HCPS case fatality rate of >40% [4,5]. ANDV is endemic in South America, especially Argentina, Chile, and Uruguay [6]. The primary reservoir is the long-tailed pygmy rat, *Oligoryzomys longicaudatus*. Hantavirus transmission to humans usually occurs through inhalation of infectious materials from urine, feces, and saliva [7], but ANDV is the only hantavirus for which human–human transmission has been reported [5,8]. A lethal hamster disease model has been established for ANDV infections closely mimicking severe human HCPS and therefore, instrumental for countermeasure development [9]. Of all human pathogenic hantaviruses, a vaccine is of particular public health interest for ANDV. Vaccination could be targeted towards certain at-risk populations or administered in the form of an emergency vaccination strategy, targeting family clusters of HCPS or wider spread outbreaks.

In a previous study we examined the potency of a live-attenuated vesicular stomatitis virus (VSV)-based vaccine vector expressing the ANDV glycoprotein precursor (GPC) in place of the VSV glycoprotein (G) (designated VSV–ANDV) in the Syrian hamster model [10]. We demonstrated the strong protection of hamsters against lethal ANDV challenge when administered pre- and peri-exposure and weaker protection when given post-exposure [10]. Surprisingly, a similar VSV vector expressing the Ebola virus (EBOV) glycoprotein (GP; designated VSV–EBOV) used for control vaccination, also mediated partial protection against lethal ANDV challenge when administered peri- and post-exposure [10]. Induction of strong innate immune responses, particularly through VSV–EBOV [10], led to the hypothesis that a bivalent vector expressing ANDV GPC on the backbone of VSV–EBOV would be a more potent ANDV vaccine candidate for emergency vaccination.

In this study we compared the protective efficacy of two monovalent (VSV-ANDV and VSV–EBOV) and two bivalent (designated VSV–ANDV–EBOV and VSV–EBOV–ANDV VSV vaccine vectors against lethal ANDV challenge in the hamster disease model. In addition, two routes of immunization—intraperitoneal (IP) and intranasal (IN)—were evaluated. The two bivalent vectors both mediated uniform protection when administered pre- and peri-exposure independent of the immunization route. The monovalent VSV–ANDV was similarly protective but IN immunization showed reduced efficacy when administered peri-exposure to ANDV challenge. The bivalent VSV–EBOV–ANDV was most potent in post-exposure immunization, followed by the non-specific monovalent VSV–EBOV. Surprisingly, the monovalent VSV–ANDV was least potent in post-exposure immunization. IP immunization was superior in post-exposure vaccination. Our data indicates that overall, the bivalent VSV–EBOV–ANDV vector is the strongest vaccine candidate for emergency intervention in case of ANDV infection clusters.

## 2. Materials and Methods

Cells and viruses. Vero E6 (African green monkey kidney) (in-house source; mycoplasma-free) and HEK 293T (human embryonic kidney cells from ATCC, CRL-3216; mycoplasma-free) were maintained in Dulbecco’s Modified Eagle’s Medium (DMEM) supplemented with 10% fetal bovine serum (FBS), 1% L-Glutamine, penicillin/streptomycin. BHK-21p (baby hamster kidney) cells (from ATCC, CCL-10; mycoplasma-free) were maintained in Gibco Minimum Essential Media (MEM) supplemented with 10% FBS, 10% tryptose phosphate broth (TBP), 1% L-Glutamine, penicillin/streptomycin. ANDV, strain Chile 9,717,869 (kindly provided by the U.S. Army Medical Research Institute of Infectious Diseases, Fort Detrick, Frederick, MD, USA), was grown and tittered on Vero E6 cells as previously described.

Generation of VSV vectors. The generation, growth, and titration of VSV vectors expressing the EBOV GP (VSV–EBOV), the ANDV GPC (VSV–ANDV) or both (VSV–ANDV–EBOV) has been described elsewhere [10,11,12]. The EBOV GP and ANDV GPC are expressed from different transcription units using VSV initiation and termination sequences, and the schematic representation of the recombinant VSVs is shown in Figure 1A. For the generation of VSV–EBOV–ANDV, the open reading frame of the ANDV GPC was amplified by iProof High-Fidelity DNA Polymerase (Bio-Rad) from plasmid DNA using the primers 5′-GTA*GTCGAC*CACCATGGAAGGGTGGTATCTGGTTGCTCTTGG-3′ (XhoI-compatible SalI restriction site underlined) and 5′-TAC*GCTAGC*TACCTATTAGACAGTTTTCTGTGTCCTCTCCTGGG-3′ (NheI restriction site underlined) (IDT). The PCR product was digested with Xhol and Nhel, purified using a PCR purification kit (Qiagen, Hilden, Germany) and cloned into the pATX–VSVΔG–EBOV plasmid downstream of the EBOV GP open reading frame [12] (Figure 1A). The resulting plasmid, pATX–VSVΔG–EBOV–ANDV, was sequence-confirmed prior to use. VSV–EBOV–ANDV (Figure 1A) was recovered as described previously [12]. A supernatant (500 µL) of cells showing cytopathogenic effect (CPE) was transferred onto fresh Vero E6 cells to generate a seed stock. The seed stock was sequenced for verification.

VSV infectivity assays. VSV vectors were titrated either using a standard plaque forming unit (PFU) assay (PFU) (seed stock titration) or a median tissue culture infectious dose (TCID_50_) assay (growth curves) as described previously.

In vitro growth of VSV vectors. Vero E6 and BHK-21 cells were grown to 90% confluence (6-well plates) and triplicates were inoculated with the VSV vectors at a multiplicity of infection (MOI) of 0.001 for 1 h at 37 °C. Subsequently, the inoculum was replaced with fresh DMEM supplemented with 3% FBS. At predetermined time points (4, 8, 12, 24, 36, 48, and 65 h post infection) 250 µL supernatant was collected from each of the triplicate wells (replaced with fresh media) and frozen at −80 °C. For titration, samples were serially diluted 10-fold and the dilutions were used to inoculate 90% confluent Vero E6 cells (triplicates) for 1 h at 37 °C. Subsequently, the inoculum was removed and replaced with fresh DMEM supplemented with 3% FBS. After 4 days of incubation at 37 °C wells were analyzed by light microscopy for CPE. The resulting TCID_50_ values were calculated using the Reed-Muench method [13].

In vivo attenuation studies. Syrian hamsters (female, 4–6 weeks of age; Charles River) (n = 4 per group) were anesthetized and IP inoculated with either DMEM (Mock control) or 1 × 10^5^ plaque forming units (PFU) (in 0.4 mL total volume of serum free DMEM) the different VSV vectors (VSV–ANDV, VSV–EBOV, VSV–ANDV–EBOV, VSV–EBOV–ANDV, and wildtype VSV (VSVwt)). Hamsters were monitored daily for adverse effects, disease progression and survival until 14 days post-infection when the animals were euthanized.

Vaccine efficacy studies. Syrian hamsters (n = 9 per group; female, 5–6 weeks of age; Charles River) were immunized IP or IN with a single dose of 1 × 10^5^ PFU of each of the VSV vectors. A single dose of VSV vectors (1 × 10^5^ PFU) was administered by the IP or IN route (0.4 mL total volume and 0.1 mL total volume, respectively) to separate cohorts either 28 days or 1 day prior to or 1, 3, or 5 days post ANDV challenge. Syrian hamsters were challenged IP (0.4 mL total volume) with a lethal dose of ANDV (200 focus forming units (FFU)) Three animals of each group were euthanized 8 days post challenge (DPC) for viral load determination in blood and lung tissue, the main target organ for ANDV replication [9,10]. The remaining 6 animals were monitored for survival for up to 40 DPC. For the post-exposure experiment an equivalent group of hamsters treated with DMEM alone was included as a negative control group that served as a control group for all experiments to save animals. The humane endpoint criteria were as follows: ataxia, extreme lethargy (reluctant to move when prompted), bleeding from any orifice, tachypnea, dyspnea, or paralysis.

Viral load determination. Lung tissue (100 mg) was placed in 1 mL of RNAlater buffer (Qiagen) overnight at 4 °C, after which it was mechanically homogenized in 600 μL of RLT lysis buffer (Qiagen), clarified by low-speed centrifugation, and then diluted to 30-mg equivalents with RLT buffer. Cardiac blood (140 μL) was mixed with 560 μL lysis buffer AVL (Qiagen). Extraction of total RNA was accomplished using the RNeasy (solid tissue) or QIAamp (blood) extraction kits (Qiagen). Quantitative real-time one-step reverse transcription-PCR (RT-PCR) was conducted on RNA extracts using a Rotor-Gene 6000 instrument (Corbett Life Science) using Quantifast Rotor-Gene Probe RT-PCR Kit reagents (Qiagen). Primers/probe targeting the ANDV nucleocapsid protein gene were: ANDV-N sense (5′-AAGGCAGTGGAGGTGGAC-3′), ANDV-N antisense (5′-CCCTGTTGGATCAACTGGTT-3′), ANDV-N probe (5′-6FAM ACGGGCAGCTGTGTCTACATTGGA-BBQ-3′) (TIB Molbiol). Primers/probe and qRT-PCR components were used at the concentrations recommended by the manufacturers. Using an in-house protocol, RNA (5 uL) was added to each reaction and the following thermocycling conditions were used: 50 °C for 10 min, 95 °C for 5 min, and 40 cycles of 95 °C for 15 s, 58 °C for 45 s. Dilutions of RNA extracted from a known titer of ANDV were run in triplicate to generate a standard curve.

Ethics and biosafety. The Institutional Biosafety Committee at Rocky Mountain Laboratories approved all standard operating protocols for work with infectious VSV (BSL2) and ANDV (tissue culture work, BSL3; animal work, BSL4). Animal work was approved by the Institutional Animal Care and Use Committee and performed by certified staff in an Association for Assessment and Accreditation of Laboratory Animal Care International accredited facility. Work followed the institution’s guidelines for animal use, the guidelines and basic principles in the NIH Guide for the Care and Use of Laboratory Animals, the Animal Welfare Act, United States Department of Agriculture and the United States Public Health Service Policy on Humane Care and Use of Laboratory Animals. Syrian hamsters were group-housed in HEPA-filtered cage systems enriched with nesting material and were provided with commercial chow and water ad libitum. Animals were monitored at least twice daily. Trained personnel carried out all procedures under isoflurane anesthesia.

Statistical analysis. Significance in survival with the VSV vectors was determined using a log-rank test (*p* < 0.01 = **). Group numbers of 6 animals per vaccine group were selected to achieve a statistical significance with a minimum statistical power of 80%. Significance was determined using a two-way ANOVA with Tukey’s multiple comparisons test [14].

## 3. Results

### 3.1. Attenuation of Monovalent and Bivalent VSV Vectors

Construction and rescue of the bivalent vaccine vector VSV–EBOV–ANDV (Figure 1A) was performed as outlined in Materials and Methods and previously described [12]. Transfected cells exhibited CPE after 12 to 14 days indicating successful recovery of VSV–EBOV–ANDV. A seed stock (passage 2) was prepared by infecting Vero E6 cells to yield a titer of 4 × 10^6^ PFU/mL as determined by standard plaque assay. Sequence confirmation revealed that no mutations arose during recovery and cell culture passage. The other VSV vectors (Figure 1A), VSV-EBOV (expressing EBOV GP), VSV-ANDV (expressing ANDV GPC) and VSV-ANDV-EBOV (expressing ANDV GPC and EBOV GP) have been described elsewhere [11].

As our vaccine efficacy model is the Syrian hamster, in vitro attenuation was analyzed on BHK-21 cells. For this, BHK-21 cells were infected with VSVwt, VSV-EBOV, VSV-ANDV, VSV-ANDV-EBOV, and VSV-EBOV-ANDV at a MOI of 0.001 and supernatants were collected at 4, 8, 12, 24, 36, 48 and 65 h post infection. All monovalent and bivalent VSV vectors replicated similarly with only minor differences among each other (Figure 1B). All were attenuated in growth kinetics compared to recombinant VSVwt, especially over the first 24 h with up to 4 log10 differences in titers (Figure 1B). VSVwt replication peaked early (12 h post infection). The monovalent and bivalent VSV gradually increased in their growth kinetics until 36 h post infection when they reached peak titers followed by a slow decline in virus titers thereafter (Figure 1B). Virus growth kinetics for all VSVs were repeated in Vero E6 cells resulting in similar replication kinetics. Altogether, the replacement of the VSV G by a single or two foreign glycoproteins resulted in in vitro attenuation with no significant difference between monovalent or bivalent VSV vectors.

To assess in vivo attenuation we used adult Syrian hamsters which are highly susceptible to VSVwt (serotype Indiana) infection [15]. Hamsters (n = 8 per group) were infected by the IP route with VSVwt, VSV-EBOV, VSV-ANDV, VSV-ANDV-EBOV or VSV-EBOV-ANDV at a dose of 1 × 10^5^ PFU per animal. All VSVwt-inoculated animals developed clinical signs of disease within 3 days of infection resulting in 50% lethality (4/8 hamsters). Except for insignificant weight loss in a few animals, hamsters in groups infected with the different VSV vectors remained asymptomatic resulting in complete survival (Figure 1C). Thus, compared to VSVwt, all monovalent and bivalent recombinant VSV vectors were attenuated in the Syrian hamster.

### 3.2. Design of the Efficacy Studies with Monovalent and Bivalent VSV Vectors

For protective efficacy testing, Syrian hamsters were immunized IP or IN either 28 days (prophylactic vaccination) or 3 days (peri-exposure vaccination) prior or 1, 3, or 5 days (post-exposure vaccination) post ANDV challenge (Figure 2A). We used a single mock control group for each immunization route which were mock vaccinated with DMEM together with the post-exposure group (D + 1) to save animals. Consequently, the virus load data in blood and lung tissue of mock control animals are the same for each time point shown in Figure 2. None of the mock immunized hamsters were protected from ANDV disease showing high viremia and ANDV lung loads and succumbed between 9 to 11 DPC (Figure 2A–G; Table 1).

### 3.3. Prophylactic Vaccination with Monovalent and Bivalent VSV Vectors

To test the efficacy of the VSV vectors as a prophylactic vaccine, hamsters were immunized 28 days prior to ANDV challenge (Figure 2A). All animals immunized with the monovalent VSV–EBOV vector by the IN route developed similar high viremia and lung loads as the mock immunized animals and succumbed to ANDV challenge within 6 to 11 DPC (Figure 2C; Table 1). Interestingly, hamsters immunized with monovalent VSV–EBOV by the IP route showed lower ANDV loads than the mock immunized and IN VSV–EBOV immunized animals, but all still succumbed to ANDV challenge within 6 to 11 DPC (Figure 2B; Table 1). All animals immunized IN with the monovalent VSV–ANDV vector were completely protected with very low or undetectable levels of ANDV genome copies in blood and lung tissue (Figure 2B; Table 1). Similarly, when animals were immunized IP almost complete protection (5 of 6) was observed with undetectable ANDV viremia and lung loads (Figure 2C; Table 1). All animals immunized with the two bivalent vectors, VSV–ANDV–EBOV and VSV–EBOV–ANDV, were completely protected, regardless of route of vaccination (Table 1) with no or very low level of ANDV genome copies in blood and lung tissue in those animals that were IN or IP immunized, respectively (Figure 2B,C). Overall, the monovalent and bivalent VSV vectors expressing the ANDV GPC were equally effective in prophylactic vaccination.

### 3.4. Rapid Vaccination with Monovalent and Bivalent VSV Vectors

To address the potential for emergency vaccination, we next immunized animals (n = 9 per group) with a single dose of 1 × 10^5^ PFU of each VSV vectors 3 days prior to lethal ANDV challenge (200 FFU/animal) (Figure 2A). Unexpectedly, hamsters IN immunized with the monovalent VSV-ANDV vector displayed moderate to high levels of ANDV genome copies in blood and lung on 8 DPC and were only partially protected (2/6; 33%) (Figure 2E; Table 1). In contrast, IP immunized hamsters had very low levels of genome copies in blood and lung and were fully protected (6/6; 100%) against lethal ANDV challenge (Figure 2D; Table 1). Interestingly, IP immunized hamsters with the monovalent VSV-EBOV vector showed 33% (2/6; 33%) protection with high levels of ANDV genome copy numbers in blood and lung (Figure 2D; Table 1), whereas IN immunization hamsters failed in protection with slightly lower levels of genome copies (Figure 2E; Table 1). All animals immunized by either route with the two bivalent vectors were uniformly protected against lethal ANDV challenge with no ANDV genome RNA detectable (Figure 2D,E; Table 1). Overall, the two bivalent VSV vectors performed better than the monovalent VSV-ANDV vector when administered IN close to challenge.

### 3.5. Post-Exposure Vaccination with Monovalent and Bivalent VSV Vectors

For post-exposure efficacy testing of the VSV vectors, hamsters (n = 9 per group) were treated IP or IN with a single dose of 1 × 10^5^ PFU on 1 day after lethal ANDV challenge (200 FFU) (Figure 2A). All hamsters IP treated with the different VSV vectors showed lower levels of ANDV genome copies on 8 DPC compared to the DMEM controls (Figure 2F). This was reflected by outcome with uniform protection for those animals treated with VSV–EBOV, VSV–EBOV–ANDV or VSV–ANDV–EBOV. Interestingly, only minimal protection was observed for VSV–ANDV (2/6; 33%) reflecting the highest viral loads among all VSV treated hamster groups (Figure 2F; Table 1). Hamsters treated by the IN route showed less protection except for hamsters IN treated with VSV–EBOV–ANDV (Figure 2G). As with the IP treatment, the monovalent VSV–ANDV vector did not perform well and showed no protection (Table 1). This outcome was reflected by high levels of ANDV genome copy detected in blood and lung in the VSV–ANDV treatment group indistinguishable from the DMEM control group (Figure 2G). The VSV–EBOV and VSV–EBOV–ANDV animals were similarly protected as the corresponding IP treatment groups, whereas the VSV–ANDV–EBOV IN treatment was less efficacious than the corresponding IP treatment (Figure 2F,G; Table 1). The best performance in post-exposure vaccination was achieved with the two bivalent vectors, with VSV–EBOV–ANDV being superior in efficacy achieving 100% (6/6) survival when given IP or IN (Table 1). Interestingly, the monovalent, non-specific VSV–EBOV vector was almost as efficacious as the bivalent vectors against ANDV challenge if administered on 1 DPC (Figure 2F,G; Table 1). Surprisingly, the monovalent specific VSV–ANDV vector performed poorly in post-exposure vaccination independent of treatment route. In general, IP treatment seemed more efficacious than IN post-exposure vaccination (Figure 2F,G; Table 1).

Finally, we tested delayed IP and IN treatment but only monitored animals (n = 6 per group) for survival in these experiments as we expected to quickly lose efficacy (Figure 2A). Surprisingly, IP immunization on 3 DPC still mediated partial protection which was highest with VSV–EBOV–ANDV (5/6; 83%) followed by VSV–EBOV (4/6; 67%) and VSV–ANDV–EBOV (1/6; 17%) (Table 1). As expected, IP immunization on 5 DPC resulted in hamsters succumbing to ANDV challenge like DMEM control animals. In contrast, IN immunization on 3 DPC and 5 DPC with any of the VSV vectors completely failed to protect hamsters against lethal ANDV challenge. Again, this shows that IP treatment with VSV vectors seemed superior to IN administration with the bivalent VSV–EBOV–ANDV performing best.

## 4. Discussion

Although HCPS outbreaks caused by ANDV are sporadic and the annual case rate is lower in South America when compared to incidences of HFRS in Europe and Asia, ANDV remains a public health concern [16]. The higher case fatality rate and the potential for human–human transmission illustrates the importance of countermeasure development including treatment options and vaccines [17]. Emergency vaccination is an effective approach to interrupt human–human transmission, as has been successfully demonstrated during EBOV outbreaks using a live-attenuated VSV-based vaccine [18] that now is licensed (Merck, Ervebo) and utilized [19]. Previously, we developed a VSV-based ANDV vaccine and demonstrated its protective efficacy in the Syrian hamster disease model [10]. The aim of this study was to improve the VSV-based vaccine to generate second-generation vectors with potent rapid onset of immunity, a prerequisite for emergency vaccination.

Formerly, VSV vectors have been extensively used as laboratory tools for studying virus and cell biology, but more recently its potential as a therapeutic and vaccine tool has been realized [20,21]. Here, we used monovalent and bivalent VSV vectors that express EBOV GP and ANDV GPC in different constellations in place of the native VSV G to determine their protective efficacy in the ANDV hamster disease model. The rationale for the bivalent vaccine vectors was based on the strong protective efficacy of the VSV–ANDV vector in prophylactic vaccination and the induction of strong innate immune responses by the VSV-EBOV when used in peri-exposure vaccination [10,22,23]. Furthermore, the favorable immune cell targeting of VSV–EBOV and the satisfactory safety profile of the vaccine vector in preclinical and clinical studies leading to its licensure as an EBOV vaccine (Merck; Ervebo; [19]) were additional strong arguments. We hypothesized that a bivalent VSV expressing ANDV GPC on the backbone of VSV–EBOV would be more potent emergency vaccines than the monovalent VSV–ANDV.

For our study we utilized two previously generated monovalent VSV vectors, VSV–EBOV and VSV–ANDV [10,12] and one bivalent vector, VSV–ANDV–EBOV [11]. To investigate the influence of the position of the foreign genes in the VSV genome, we generated a new bivalent VSV vector, VSV–EBOV–ANDV (Figure 1A). To test vaccine safety and efficacy we used the Syrian hamster model. Although ANDV results in systemic infection in hamsters, the lungs are the primary site of viral replication and the only organ which develops pathological abnormalities over the course of infection [9,24]. While VSVwt also causes systemic infection in hamsters that may result in severe and partially lethal outcomes, the VSV-based vaccine candidates generated here were attenuated in vitro (hamster cells) and in vivo (Syrian hamsters) (Figure 1B,C).

We predicted that, like IP immunization, administration of the ANDV containing VSV vaccines via the IN route would provide strong protection when given 28 days prior to exposure. This was indeed the case for the monovalent VSV-ANDV and the two bivalent VSV vectors with no significant difference in ANDV loads and survival (Figure 2B,C; Table 1). We expected similar results for the peri-exposure vaccination, which was the case for the IP route, but IN immunization with the monovalent VSV-ANDV clearly showed breakthrough in ANDV load and survival indicating lower efficacy (Figure 2D,E; Table 1). The two bivalent vectors, however, showed potent protection when IN immunization occurred peri-exposure. While we hypothesized that IN immunization post-exposure would provide increased efficacy by eliciting a strong innate followed by an effective adaptive immune response in the respiratory tract of the hamster, we experienced superiority of the IP immunization route (Figure 2F,G; Table 1). One explanation for this outcome may be a quantitative difference in the amount of vaccine delivered by the two routes of vaccination. The IP route delivers a known dose with high certainty, whereas this may not be the case with the IN route as the uptake of the vaccine through the mucosal surface is unlikely to be complete. Furthermore, IN vaccination may be more effective in the peri-exposure setting in case of mucosal ANDV challenge. The bivalent VSV vectors showed better protection than the monovalent VSV–ANDV which basically failed in post-exposure prophylaxis against ANDV challenge. This contrasts with a previous EBOV challenge study were the monovalent VSV–EBOV had a slightly increased postexposure efficacy over the bivalent VSV–ANDV–EBOV vector most likely due to its restricted lymphoid organ replication [11]. Interestingly, in our study here the monovalent, non-specific VSV–EBOV was as potent in protection as the bivalent vectors in post-exposure prophylaxis (Figure 2F,G; Table 1). This supports the choice of the bivalent VSV vectors and indicates that unspecific innate immune responses are key to protection in this scenario as has been previously hypothesized [10,22,23].

Looking at the viral load data, the monovalent VSV–ANDV and the two bivalent VSV vectors strongly reduced or even completely prevented ANDV replication following IN and IP immunization pre- and peri-exposure (Figure 2B–E). Thus, any of those VSV vectors could be moved forward along the pathway to develop an ANDV vaccine. All VSV vectors here, however, failed in preventing ANDV replication when administered post-exposure despite protection from ANDV associated disease and lethality. This is not surprising, but it could lead to problems such as acceleration of virus escape mutant development, as has been shown for other viruses [25], or ANDV persistence in immune privileged sites as has been described for infected human cases [26]. These are problems that need to be further investigated.

Despite a benefit for survival with especially the bivalent VSV vectors when administered early post exposure (1 DPC), the relatively quick loss of efficacy when administered delayed (3 DPC and 5 DPC) questions the use of VSV vectors as a treatment option for ANDV infections. Here, treatment with monoclonal antibodies (MAB) seems more effective showing preclinical efficacy in mid- or late-stage disease in the hamster model [27,28,29,30,31]. Thus, MAB treatment may be a more feasible and valuable approach for clinical management of HCPS patients. However, combination treatment consisting of post-exposure VSV vector vaccination, especially the bivalent VSV vectors, and MAB treatment could be evaluated for increased efficacy. The MABs directed to the ANDV GPC would unlikely interfere with the bivalent VSV vectors as they would not neutralize the EBOV GP which is likely the main driver of cell attachment and membrane fusion of the VSV vectors. In addition, for VSV–EBOV, it was shown that prior vaccination of nonhuman primates did not just interfere but even improved subsequent MAB treatment [32].

As this study supports a stronger potency of the bivalent VSV vectors over the monovalent VSV–ANDV, the position of the ANDV GPC and EBOV GP genes in the genome of the bivalent constructs becomes a relevant question. Overall, both bivalent VSV vectors performed similarly but with post-exposure vaccination increased efficacy was associated with VSV–EBOV–ANDV. Due to the steepened 3′ to 5′ transcription gradient of VSV genes [33], this may suggest increased EBOV GP production with VSV–EBOV–ANDV over VSV–ANDV–EBOV leading to a stronger innate immune response something that would have to be further investigated.

Our study has several limitations and future studies will have to augment the findings presented here. First, we focused on virus load and survival as parameters for protective efficacy, but did not correlate this with the hamster immune responses which, based on previous work, would be largely antibodies [10]. The main purpose of this study, however, was to identify the most protective VSV vector and vaccination route with focus on survival. Second, we did not determine the protective efficacy against EBOV challenge. As we already know, contrary to VSV–ANDV, VSV–EBOV and VSV–ANDV–EBOV protected against lethal EBOV challenge in the mouse model [10,11], we did not perform EBOV challenge studies here, predicting that VSV–EBOV–ANDV would be as protective. Third, we did not determine innate responses likely initially responsible for peri- and post-exposure immunization. Despite establishing workable assays earlier [34], preliminary data in this study demonstrated the need for further optimization prior to use.

## 5. Conclusions

Here we compared the protective efficacy of two monovalent and two bivalent VSV vectors against lethal ANDV challenge in the hamster disease model. Our data indicates that VSV–EBOV–ANDV is the superior vaccine candidate for emergency vaccination in case of ANDV infection clusters to prevent human–human transmission.

## Figures and Tables

**Figure 1 viruses-16-00279-f001:**
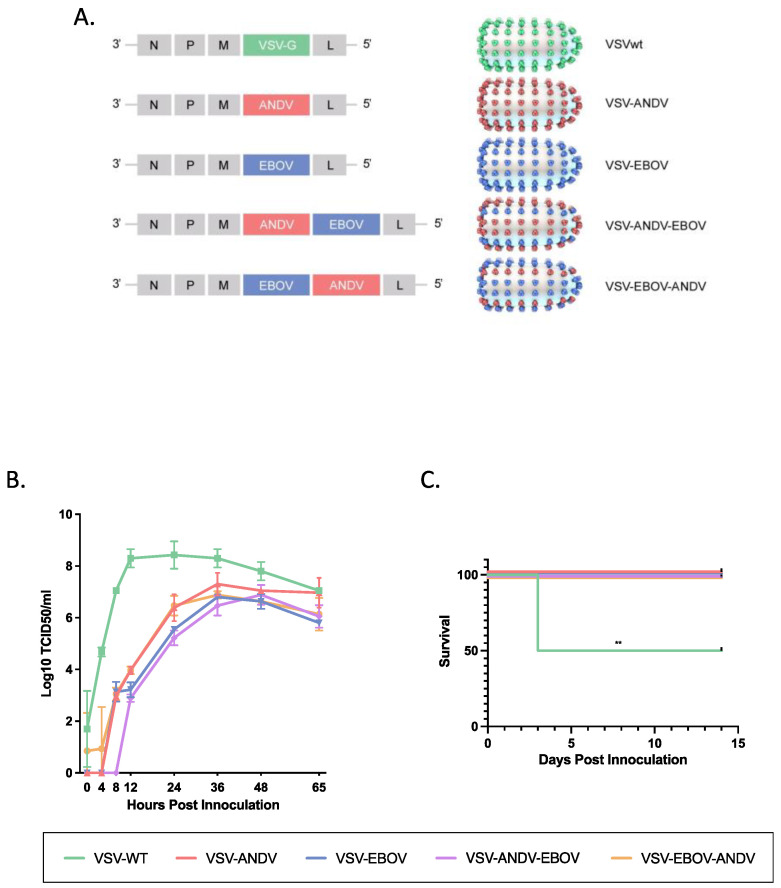
Generation and characterization of the VSV vectors. (**A**) Schematic design of VSV vectors. VSV vectors were generated through reverse genetics, recovered, titrated and sequence confirmed. The following VSV vectors were used: VSVwt, VSV–ANDV (expressing ANDV GPC), VSV–EBOV (expressing EBOV GP), VSV–ANDV–EBOV (expressing ANDV GPC and EBOV GP), and VSV–EBOV–ANDV (expressing EBOV GP and ANDV GPC). (**B**) In vitro attenuation. BHK-21 cells were infected with the different VSV vectors (MOI of 0.001), and supernatants were harvested at the indicated time points. Infectivity was determined by a TCID_50_ assay. (**C**) In vivo attenuation. Syrian hamsters (*n* = 8) were infected with the different VSV vectors and monitored for clinical signs. The graph shows the survival curves. A log-rank test was used to determine significance (*p* < 0.01 = **). ANDV = Andes virus; EBOV = Ebola virus; MOI = multiplicity of infection; TCID50 = median tissue culture infectious dose.

**Figure 2 viruses-16-00279-f002:**
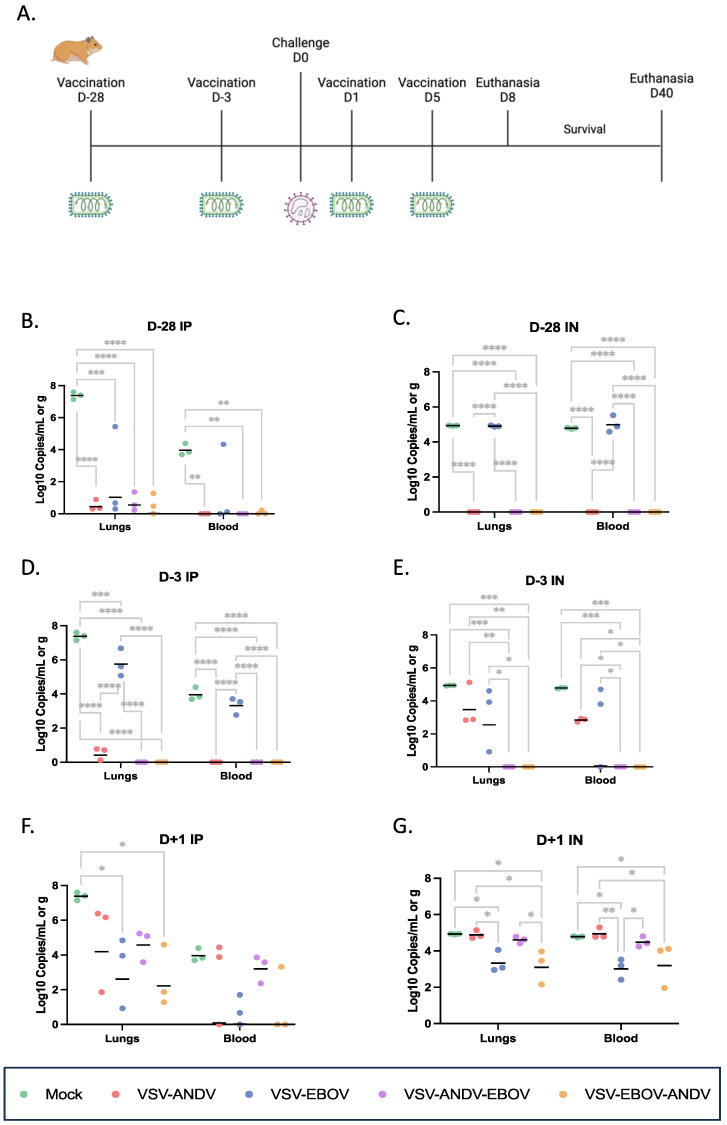
Viral load in lung tissue and blood after pre- and post-challenge immunization. (**A**) Experimental design. Hamsters (*n* = 9 per group) were immunized IP or IN with a single dose of 1 × 10^5^ PFU pre (D-28 and D-3) or post (D + 1) lethal ANDV challenge (200 FFU/animal). Lung tissue and blood were taken from 3 animals of each group on 8 DPC and analyzed for viral replication using RT-PCR. (**B**) D-28, IP immunization; (**C**) D-28, IN immunization; (**D**) D-3, IP immunization; (**E**) D-3, IN immunization; (**F**) D + 1, IP immunization; (**G**) D + 1, IN immunization. Significance between groups was determined using a two-way ANOVA with Tukey’s multiple comparisons test (*p* < 0.0001 = ****, *p* < 0.001 = ***, *p* < 0.01 = **, *p* < 0.05 = *). ANDV = Andes virus; D = day; EBOV = Ebola virus; IN = intranasal; IP = intraperitoneal; ‘-’ = pre challenge; ‘+’ = post challenge.

**Table 1 viruses-16-00279-t001:** Survival from ANDV challenge following immunization with VSV vectors.

	D-28 No. (%)	D-3 No. (%)	D + 1 No. (%)	D + 3 No. (%)	D + 5 No. (%)
VSV–EBOV IP	0/6 (0)	2/6 (33)	6/6 (100)	4/6 (67)	6/6 (0)
VSV–EBOV IN	0/6 (0)	0/6 (0)	5/6 (83)	0/6 (0)	0/6 (0)
VSV–ANDV IP	5/6 (83)	6/6 (100)	2/6 (33)	0/6 (0)	1/6 (17)
VSV–ANDV IN	6/6 (100)	2/6 (33)	0/6 (0)	0/6 (0)	0/6 (0)
VSV–ANDV–EBOV IP	6/6 (100)	6/6 (100)	6/6 (100)	1/6 (17)	0/6 (0)
VSV–ANDV–EBOV IN	6/6 (100)	6/6 (100)	3/6 (50)	0/6 (0)	0/6 (0)
VSV–EBOV–ANDV IP	6/6 (100)	6/6 (100)	6/6 (100)	5/6 (83)	0/6 (0)
VSV–EBOV–ANDV IN	6/6 (100)	6/6 (100)	6/6 (100)	0/6 (0)	0/6 (0)
Mock (DMEM) IP	ND	ND	0/6 (0)	ND	ND
Mock (DMEM) IN	ND	ND	0/6 (0)	ND	ND

Hamsters (n = 6 per group) were immunized IP or IN with a single dose of 1 × 10^5^ PFU of each VSV vectors pre (D-28; D-3) and post (D + 1; D + 5) lethal ANDV challenge (200 FFU/animal/IP). Animals were monitored daily for survival up to 40 days post-challenge. ANDV = Andes virus; D = day; DMEM = Dulbecco’s Modified Eagle’s Medium EBOV = Ebola virus; IN = intranasal; IP = intraperitoneal; no. = survived/deceased; ‘-’ = pre challenge; ‘+’ = post challenge; % = percentage survival).

## Data Availability

All analyzed data are available in this manuscript. Raw data will be provided upon request from the corresponding authors.

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
