# Peer review of "Bivalent VSV Vectors Mediate Rapid and Potent Protection from Andes Virus Challenge in Hamsters"

_viruses, 2024, doi:10.3390/v16020279_

Round 1
Reviewer 1 Report
Comments and Suggestions for Authors
Very well constructed and performed experimental design. This is a logical next step to earlier work that suggested efficacy of a VSV-EBOV-ANDV virus bivalent construct would be an improvement on promising approaches to enhance protective efficacy. Results did present some intriguing findings related to the inherent value of innate immune responses in contributing to partial protective effects (EBOV up regulation of those responses).
Two points for the authors to consider.
I would suggest adding a line at the end of section 2.6 to help clarify a slight confusion related to how many post-challenge vaccination were scheduled. Something along the line of "A single dose of vaccine was administered to separate cohorts either 28 days or 1 day pre-challenge, or 1, 3 or 5 days post-challenge with Andes Virus." Otherwise,it was not apparent to me in my reading that there were post-challenge doses administered on days 3 or 5. My first understanding of this happened in looking at Table 1.
Second question / comment. Could there have been a quantitative difference in the amount of vaccine delivered by IP vs. IN routes? The IP route offers the certainty that a known dose is provided. The IN route is less certain in this regard. Could there have been a difference in antigenic mass presented to the subject animals which may have had a potential for the IN administration to not have been as potent a dose as the IP?
Author Response
Reviewer 1
Very well constructed and performed experimental design. This is a logical next step to earlier work that suggested efficacy of a VSV-EBOV-ANDV virus bivalent construct would be an improvement on promising approaches to enhance protective efficacy. Results did present some intriguing findings related to the inherent value of innate immune responses in contributing to partial protective effects (EBOV up regulation of those responses).
We appreciate the positive assessment by the reviewer.
Two points for the authors to consider.
I would suggest adding a line at the end of section 2.6 to help clarify a slight confusion related to how many post-challenge vaccinations were scheduled. Something along the line of "A single dose of vaccine was administered to separate cohorts either 28 days or 1 day pre-challenge, or 1-, 3- or 5-days post-challenge with Andes Virus." Otherwise, it was not apparent to me in my reading that there were post-challenge doses administered on days 3 or 5. My first understanding of this happened in looking at Table 1.
Thanks for the valuable comment. A sentence has been added (lines 132-135).
Second question / comment. Could there have been a quantitative difference in the amount of vaccine delivered by IP vs. IN routes? The IP route offers the certainty that a known dose is provided. The IN route is less certain in this regard. Could there have been a difference in antigenic mass presented to the subject animals which may have had a potential for the IN administration to not have been as potent a dose as the IP?
This is an interesting thought, but we don’t have the answer. We added this thought to the Discussion section (lines 400-404).
Reviewer 2 Report
Comments and Suggestions for Authors
1-In the Introduction section:
This a well elaborated part of this article and the authors have done a good effort in forwarding the rationale and specific objectives of this experimental research but in my view in page 2 line 45, the appropriate references may be 8 and 9 rather than 7 and 8 because the authors referred to thier "previous studies" please check check for that.Moreover it would have been more informative if enough data is included about the epidemiology of Andes virus recently and in the past .
2-The Materials and methods section:
Most of the protocols needed to meet a good standard of such applied research were followed .but for 2.2 Generation of VSV vectors ,2.5. In vivo attenuation studies and 2.6. Vaccine efficacy studies it would be of benefit to readers if references were provided plus the source of the primers used in the molecular analysis are they in house made or commercially availabe?.
3-In the results section:
3.2. Design of the efficacy studies with monovalent and bivalent VSV vectors page5 lines 211-216 this will be most appropriate to be placed in the methodology section only the results to be shown here
page 7 line 287 the pre challenge and post challenge (‘-‘ = pre challenge; ‘+’ = post challenge) were not shown in table (1) and were described below it .
4-The discussion section:
The data was adequately presented according to the results findings and gives evidence based data about the advantage of using bivalent VSV vector vaccines in comparison to monovalent vaccines and how it could be of benefit if given earlier in course of the disease but in my opinion this needs to be a separate research effort that will augiment the present findings .The authors have honestly mentioned several limitations of the present study which may be overcomed in future studies .
My overall view this study will be a good example for the future use of VSV vector vaccines for highly pathogenic viruses and may be of benefit to use in other infectious and non infectious disease models.
Author Response
Reviewer 2
1-In the Introduction section:
This a well elaborated part of this article and the authors have done a good effort in forwarding the rationale and specific objectives of this experimental research but in my view in page 2 line 45, the appropriate references may be 8 and 9 rather than 7 and 8 because the authors referred to their "previous studies" please check for that. Moreover it would have been more informative if enough data is included about the epidemiology of Andes virus recently and in the past.
We did check and correct the references (line 59). We also added more recent and past information on the epidemiology of Andes virus (lines 44-55)
2-The Materials and methods section:
Most of the protocols needed to meet a good standard of such applied research were followed but for 2.2 Generation of VSV vectors ,2.5. In vivo attenuation studies and 2.6. Vaccine efficacy studies it would be of benefit to readers if references were provided plus the source of the primers used in the molecular analysis are they in house made or commercially available?
We have added references to those sections and added the source of the primers (line 101; line 166).
3-In the results section:
3.2. Design of the efficacy studies with monovalent and bivalent VSV vectors page5 lines 211-216 this will be most appropriate to be placed in the methodology section only the results to be shown here.
We have moved this part to the Method section (lines 130-153).
page 7 line 287 the pre challenge and post challenge (‘- ‘= pre challenge; ‘+’ = post challenge) were not shown in table (1) and were described below it.
The symbols ‘- ‘and ‘+’ referring to the days prior to or post challenge. Those symbols are used in the figures as well as Table 1 (upper row; for example, D-28 or D+5).
4-The discussion section:
The data was adequately presented according to the results findings and gives evidence based data about the advantage of using bivalent VSV vector vaccines in comparison to monovalent vaccines and how it could be of benefit if given earlier in course of the disease but in my opinion this needs to be a separate research effort that will augment the present findings .The authors have honestly mentioned several limitations of the present study which may be overcome in future studies .
We have added a comment on future separate efforts to augment the present findings (lines 446-447).
My overall view this study will be a good example for the future use of VSV vector vaccines for highly pathogenic viruses and may be of benefit to use in other infectious and non-infectious disease models.
We appreciate the positive assessment by the reviewer.
Reviewer 3 Report
Comments and Suggestions for Authors
In the present research article, Marceau et al, describe the use of vesicular stomatitis virus (VSV)-based vaccine to prevent Andes virus associated disease. Andes virus is a hantavirus which is associated to cause Hantavirus pulmonary syndrome (HPS) in humans. This virus is distributed in South-America and is the only hantavirus shown to be transmitted between humans. Until now, there is no antiviral treatment or vaccines available. Previously, the authors of this paper have developed this VSV system to express Ebola virus glycoprotein (EBOV GP) which had positive results as a vaccine. Also, the authors have described the use of VSV system for ANDV GPC (monovalent) and ANDV GPC-EBOV GP (bivalent) as vaccine in Syrian hamsters which is the model for ANDV HPS. In the present manuscript, have used all the vectors described previously and have added a new VSV system (EBOV GP-ANDV GPC) to try to obtain better results than the previously published. The authors have also added the intranasal route to deliver the viral vectors.
In general, the manuscript is well written but lacks novelty. The authors analyzed the viral loads in blood and the lung which is the primary organ of infection during the disease and according to these results, they concluded that the new bivalent vector (VSV-EBOVGP-ANDVGPC) was the best for post exposure immunization and suitable for emergency vaccination. The authors analyzed survival as a parameter to conclude that intraperitoneal route with the bivalent vector was the best for emergency vaccination. The paper has many limitations, but the main concern is that the immune response against this new bivalent vector was not addressed, therefore the conclusions obtained are not credible.
My suggestions and concerns
1-To show a scheme of the experimental design would be much helpful to follow the experiments
2- The percentage of survival shown in the table is not proper since, the authors have analyzed the survival of 6 animals per condition and in the table to put 33% is not a correct. I suggest indicating it as 2/6, etc
3- My main concern about the immune response against de VSV system in Syrian hamsters is why to use EBOV as control since is not described the Syrian hamster as a model for EBOV disease. In fact, the immune response for VSV-EBOV in Syrian hamsters is not described at all, therefore, this is something that is very limiting for the entire experimental design.
Comments on the Quality of English LanguageThe english is ok
Author Response
Reviewer 3
In the present research article, Marceau et al, describe the use of vesicular stomatitis virus (VSV)-based vaccine to prevent Andes virus associated disease. Andes virus is a hantavirus which is associated to cause Hantavirus pulmonary syndrome (HPS) in humans. This virus is distributed in South America and is the only hantavirus shown to be transmitted between humans. Until now, there is no antiviral treatment or vaccines available. Previously, the authors of this paper have developed this VSV system to express Ebola virus glycoprotein (EBOV GP) which had positive results as a vaccine. Also, the authors have described the use of VSV system for ANDV GPC (monovalent) and ANDV GPC-EBOV GP (bivalent) as vaccine in Syrian hamsters which is the model for ANDV HPS. In the present manuscript, have used all the vectors described previously and have added a new VSV system (EBOV GP-ANDV GPC) to try to obtain better results than the previously published. The authors have also added the intranasal route to deliver the viral vectors.
In general, the manuscript is well written but lacks novelty. The authors analyzed the viral loads in blood and the lung which is the primary organ of infection during the disease and according to these results, they concluded that the new bivalent vector (VSV-EBOVGP-ANDVGPC) was the best for post exposure immunization and suitable for emergency vaccination. The authors analyzed survival as a parameter to conclude that intraperitoneal route with the bivalent vector was the best for emergency vaccination. The paper has many limitations, but the main concern is that the immune response against this new bivalent vector was not addressed, therefore the conclusions obtained are not credible.
The study was designed with the intent to define the best VSV-based vector for prophylactic and post-exposure vaccination and to determine the best route of vaccination. Both aims have not been addressed in previous studies. Thus, in our view this study addresses new aspects, and the results will be helpful for future development of a VSV-based Andes virus vaccine.
We agree with the reviewer that the evaluation of immune responses would have been a helpful additional data set. Unfortunately, Andes disease in hamster is highly acute and animals go from mild/moderate clinical signs to humane endpoint in hours. We lost too many animals overnight with no samples for serology and other immunological analyses. Overall, the animal numbers with a blood draw in the different groups were too inconsistent to do meaningful serology and a comparison among groups. Therefore, we have based the assessment on virus load and survival. The Andes hamster disease model is well established in the field, and we have experience with VSV vector immunization in hamsters for multiple VSV-based vaccine vectors targeting a variety of infectious diseases caused by hantaviruses, arenaviruses, filoviruses, and paramyxoviruses. In general, immunization generates a strong humoral immune response with IgM and IgG antibodies but a weak or undetectable T cell response. We rarely have a hamster that does not respond to VSV vector immunization. Therefore, we feel that the serology is not critical for the evaluation of our work, even though it would have been a welcomed addition if feasible.
My suggestions and concerns
1-To show a scheme of the experimental design would be much helpful to follow the experiments
We have added a scheme of the experimental design as suggested by the reviewer. This is now shown in the revised Figure 2A.
2- The percentage of survival shown in the table is not proper since, the authors have analyzed the survival of 6 animals per condition and in the table to put 33% is not a correct. I suggest indicating it as 2/6, etc.
We have followed the advice of the reviewer to add the animal numbers to the table. We, however, also left the percentage in the table for easier evaluation of the results. (Table 1)
3- My main concern about the immune response against de VSV system in Syrian hamsters is why to use EBOV as control since is not described the Syrian hamster as a model for EBOV disease. In fact, the immune response for VSV-EBOV in Syrian hamsters is not described at all, therefore, this is something that is very limiting for the entire experimental design.
The Syrian hamster is a lethal Ebola disease model (Ebihara et al. J Infect Dis. 2013 Jan 15;207(2):306-18), and the model has been used by researchers in the field to study pathogenesis and develop countermeasures. We have described the immune response to VSV-EBOV in the Syrian hamsters previously (Tsuda et al. J Infect Dis. 2011 Nov;204 Suppl 3(Suppl 3):S1090-7).
Serology was not assessed here due to inconsistent sample availability from animals that succumbed overnight in those groups with partial or no protection. This deficiency would have not allowed us to produce comparable data among the different groups. Nevertheless, we think that virus load data and survival are critical elements that allow an evaluation of the different vaccine vectors (see also comment above).
Reviewer 4 Report
Comments and Suggestions for Authors
The authors continue their work describing various aspects of the VSV-vectored viral vaccines. Here, they generate a vaccine that encodes EBOV (filovirus) and ANDV (hantavirus) glycoproteins in this order to complement a previous vaccine with the proteins in the reverse order and the corresponding monovalent vaccines. They show growth curves for the recombinants, basic attenuation in hamsters as compared to wild-type VSV, as well as challenge virus titers and survival after vaccination at various time-points and using two routes. Serology or T cell data are not presented.
Although the viral load and survival data don’t quite go hand-in-hand, the picture that seems to emerge is that 1) in standard prophylactic setting, two antigens are tolerated and animals are protected equally well as with vaccine encoding ANDV GP only, and 2) presence of EBOV GP appears to provide a short window of protective (innate) immunity when the vaccine is given close to challenge. The animal experimentation is extensive and the data is of interest to the field, particularly since this and other VSV vectors are being developed against various viral pathogens.
The manuscript text is well-written and thorough, but I have minor comments:
- Materials and Methods 2.2: Please mention how the two glycoproteins are expressed from the VSV-G locus. Are these two different transcription units? Do they both use the initiation and termination sequences of VSV-G?
- Materials and Methods 2.6: Please define the humane end-points for the animals.
- Discussion: consider commenting more on the pros and cons of the IP and IN vaccine routes. Looking at table 1 and figure 2, IN seems to give tighter variance and the titers and survival are in better concordance than with IP..?
- Line 41: consider adding more references regarding ANDV human-to-human transmission. Several papers are out, and the matter is perhaps more settled than (I for one) earlier thought.
- Having seen the quite short window of innate protection conferred by the EBOV GP, consider commenting on the incubation period of ANDV HPS in humans.
- Figure 1: Please double check that the colors of the legend match the figures. It seems, on my screen at least, that there are purple graphs but no purple group.
Author Response
Reviewer 4
The authors continue their work describing various aspects of the VSV-vectored viral vaccines. Here, they generate a vaccine that encodes EBOV (filovirus) and ANDV (hantavirus) glycoproteins in this order to complement a previous vaccine with the proteins in the reverse order and the corresponding monovalent vaccines. They show growth curves for the recombinants, basic attenuation in hamsters as compared to wild-type VSV, as well as challenge virus titers and survival after vaccination at various time-points and using two routes. Serology or T cell data are not presented.
Although the viral load and survival data don’t quite go hand-in-hand, the picture that seems to emerge is that 1) in standard prophylactic setting, two antigens are tolerated and animals are protected equally well as with vaccine encoding ANDV GP only, and 2) presence of EBOV GP appears to provide a short window of protective (innate) immunity when the vaccine is given close to challenge. The animal experimentation is extensive, and the data is of interest to the field, particularly since this and other VSV vectors are being developed against various viral pathogens.
We appreciate the encouraging assessment by the reviewer.
The manuscript text is well-written and thorough, but I have minor comments:
- Materials and Methods 2.2: Please mention how the two glycoproteins are expressed from the VSV-G locus. Are these two different transcription units? Do they both use the initiation and termination sequences of VSV-G?
The requested information has been added (lines 94-95).
- Materials and Methods 2.6: Please define the humane endpoints for the animals.
The requested information has been added (lines 151-153).
- Discussion: consider commenting more on the pros and cons of the IP and IN vaccine routes. Looking at table 1 and figure 2, IN seems to give tighter variance and the titers and survival are in better concordance than with IP..?
We have added additional comments on IP and IN vaccinations (lines 400-405).
- Line 41: consider adding more references regarding ANDV human-to-human transmission. Several papers are out, and the matter is perhaps more settled than (I for one) earlier thought.
We have added more references to the topic of human-to-human transmission (lines 43-55).
- Having seen the quite short window of innate protection conferred by the EBOV GP, consider commenting on the incubation period of ANDV HPS in humans.
The incubation period of Andes virus induced HCPS varies from 7-39 days with a median of 18 days (Pablo et al., Emerg Infect Dis. 2006 Aug; 12(8): 1271–1273). Therefore, it is difficult to connect innate immune responses with disease onset. The impact of the innate immune response seems much higher on virus replication which would start within hours/day of exposure. Therefore, we chose to not add a comment to the revised manuscript.
- Figure 1: Please double check that the colors of the legend match the figures. It seems, on my screen at least, that there are purple graphs but no purple group.
Thanks for noticing the mistake which has been corrected in the revised figures.
Round 2
Reviewer 3 Report
Comments and Suggestions for Authors
In the present manuscript, the authors have made the changes to improve it. In the discussion part, line 428-429 the authors reference themselves for the use of protective mabs in hamsters but are other authors that also have made good progress in this matter that should be added as reference, Mittler, 2022 (Human antibody recognizing a quaternary epitope in the Puumala virus glycoprotein provides broad protection against orthohantaviruses) and Mittler, 2023 (Structural and mechanistic basis of neutralization by a pan-hantavirus protective antibody).
Author Response
Point-To-Point Rebuttal
Reviewer #3
In the present manuscript, the authors have made the changes to improve it. In the discussion part, line 428-429 the authors reference themselves for the use of protective mabs in hamsters but are other authors that also have made good progress in this matter that should be added as reference, Mittler, 2022 (Human antibody recognizing a quaternary epitope in the Puumala virus glycoprotein provides broad protection against orthohantaviruses) and Mittler, 2023 (Structural and mechanistic basis of neutralization by a pan-hantavirus protective antibody).
We thank the reviewer for pointing out two more publication to be cited for potential antibody therapy of hantavirus infections. We have added Mittler et al., 2022 (citation see below); the second suggestion, Mittler et al 2023, is already cited as reference #29 in revision 1.
Mittler E, Wec AZ, Tynell J, Guardado-Calvo P, Wigren-Byström J, Polanco LC, O'Brien CM, Slough MM, Abelson DM, Serris A, Sakharkar M, Pehau-Arnaudet G, Bakken RR, Geoghegan JC, Jangra RK, Keller M, Zeitlin L, Vapalahti O, Ulrich RG, Bornholdt ZA, Ahlm C, Rey FA, Dye JM, Bradfute SB, Strandin T, Herbert AS, Forsell MNE, Walker LM, Chandran K. Human antibody recognizing a quaternary epitope in the Puumala virus glycoprotein provides broad protection against orthohantaviruses. Sci Transl Med. 2022 Mar 16;14(636):eabl5399. doi: 10.1126/scitranslmed.abl5399. Epub 2022 Mar 16. PMID: 35294259; PMCID: PMC9805701.